# Long-Term Neurological Outcomes of Adult Patients with Phenylketonuria before and after Newborn Screening in Japan

**DOI:** 10.3390/ijns7020021

**Published:** 2021-04-14

**Authors:** Kenji Yamada, Seiji Yamaguchi, Kazunori Yokoyama, Kikumaro Aoki, Takeshi Taketani

**Affiliations:** 1Department of Pediatrics, Shimane University Faculty of Medicine, 89-1 Enya-cho, Izumo, Shimane 693-8501, Japan; seijiyam@med.shimane-u.ac.jp (S.Y.); ttaketani@med.shimane-u.ac.jp (T.T.); 2Secretariat of Special Formula, Aiiku Maternal and Child Health Center, Imperial Gift Foundation Boshi-Aiiku-Kai, 5-6-8 Minami Azabu, Minato-ku, Tokyo 106-8580, Japan; milk@boshiaiikukai.jp (K.Y.); aoki@boshiaiikukai.jp (K.A.)

**Keywords:** phenylketonuria, newborn screening, long-term outcome, adult patients, Japanese, intellectual disability, psychiatric disability, treatment discontinuation

## Abstract

Japanese newborn screening (NBS) for phenylketonuria (PKU) was initiated in 1977. We surveyed the neurological outcomes of Japanese adult patients with PKU to investigate the long-term effects and of and issues with NBS. Eighty-five patients with PKU aged over 19 years who continued to be treated with a phenylalanine-free amino acid formula were investigated by administering questionnaires regarding clinical characteristics, such as mental ability, education status, and therapeutic condition. Of the 85 subjects, 68 patients were detected by NBS (NBS group), while the other 17 were clinically diagnosed before the initiation of NBS (pre-NBS group). Further, 10 of the 68 NBS patients presented intellectual and/or psychiatric disabilities, 5 of whom had a history of treatment discontinuation; in contrast, 12 of the 17 pre-NBS patients presented with neuropsychiatric symptoms. Regarding social outcomes, almost all patients in the NBS group could live an independent life, while over half of the patients in the pre-NBS group were not employed or lived in nursing-care facilities. Neurological outcomes are obviously improved by NBS in Japan. However, some patients, even those detected by NBS, developed neuropsychiatric symptoms due to treatment disruption. Lifelong and strict management is essential to maintain good neurological and social prognoses for patients with PKU.

## 1. Introduction

Phenylketonuria (PKU, OMIM No. 261600), an autosomal recessive inherited disease of amino acid metabolism, is a major disease identified by newborn screening (NBS) [1]. Classic PKU is caused by phenylalanine hydroxylase (PAH, OMIM No. 612349) deficiency and is associated with severe intellectual disability, convulsions, hypopigmentation, behavioral abnormalities, and dementia. Because PKU is detected as an elevation in phenylalanine (Phe) levels in NBS, mild hyperphenylalaninemia (HPA), tetrahydrobiopterin (BH_4_) deficiency, including BH_4_ metabolic defects, and BH_4_-responsive PKU and classic PKU are all detectable. Mild HPA is also caused by a defect in PAH but is clinically mild and may require less strict dietary restrictions. BH_4_-responsive PKU and BH_4_ deficiency show BH_4_ responsiveness. The global prevalence of PKU has been reported to be approximately 1 in 4000 to 15,000 births [1,2,3], but the incidence of NBS in Japan has been reported to be 1 in 50,000 to 70,000 births [4,5].

NBS for PKU first spread from the mid to late 1960s in North America and the United Kingdom, while nationwide NBS for PKU was initiated in 1977 in Japan. Almost all cases of PKU are detected by NBS and treated immediately after birth. The treatment is mainly a diet with protein restriction and Phe-free formula and supplementation with BH_4_ in some cases. With this therapy, the life and neurological prognoses of patients with PKU have been reported to improve, at least in childhood, resulting in a significantly decreased societal economic burden [6,7]. Therefore, PKU is considered one of the best targets for NBS in terms of a cost–benefit analysis.

However, even in patients who are initially treated and achieve good metabolic control, reversible (and sometimes irreversible) neuropsychiatric symptoms can develop if this control is lost in later childhood or adulthood [8]. In addition, over time, some patients may manifest subtle intellectual and neuropsychiatric issues even with strict adherence to a low-Phe diet [9,10,11,12]. In fact, the long-term outcome of PKU is not necessarily favorable.

In this study, we expanded upon our previous studies [5,13,14] and investigated the social and neurological outcomes of adult patients aged over 20 years who were diagnosed with PKU before and after NBS in Japan using a questionnaire administered to attending doctors.

## 2. Materials and Methods

We sent questionnaires to the 33 attending physicians of 85 adult patients with PKU aged over 20 years as of October 2016 (born before 1996) who had continuously required a special formula (including Phe-free comprehensive amino acid powder and low-Phe peptide powder) supplied by the Secretariat of Special Formula, Aiiku Maternal and Child Health Center for dietary therapy.

Attending physicians answered the questionnaires based on clinical records. The questionnaire included questions on sex, age, clinical form of PKU, history of treatment interruption and the reason, follow-up department, physical development, neuropsychiatric disability, educational status, work status, and marital status and allowed for an open-ended description of the patient.

Regarding the question on neuropsychiatric disability, attending physicians were able to answer “normal intelligence”, “borderline intellectual disability”, “intellectual disability”, and/or “psychiatric disability”. When “intellectual disability” was selected, patients with an IQ (intelligence quotient) less than 50 or patients living in nursing-care facilities were characterized as having a “moderate-severe intellectual disability”, while those with an IQ of 50–70 or who are able to fend for themselves and patients with borderline intellectual disability were characterized as having a “borderline-mild intellectual disability”. Additionally, the details of psychiatric disability were judged based on the open-ended description. Regarding physical development, short stature was defined as a height <160 cm for males and <148 cm for females. Obesity and leanness were defined as BMI >25 and <18.5 kg/m^2^, respectively.

The subjects were divided into the following two groups: (1) the NBS group was defined as patients who underwent NBS, and (2) the pre-NBS group consisted of patients who were born before 1977 (when NBS was initiated) or were born after 1977 but did not receive NBS. Each of the groups was subdivided into age groups of approximately 5 years. The results of the NBS and pre-NBS groups were compared without a statistical analysis because of the small-scale nature of this study.

This study was approved by the Institutional Review Board of Shimane University on 13 October 2016 (#20160915-2).

## 3. Results

We received answers from the attending physicians of all 85 adult patients with PKU (response rate of 100%). All participants were alive. Because only one patient had not undergone NBS despite being born in 1979, she was enrolled in the pre-NBS group (and classified into the 39–44-year-old subgroup). Eventually, 68 and 17 patients were placed into the NBS and pre-NBS groups, respectively. As shown in Table 1, the NBS group consisted of 34 males and 33 females, while 7 males and 10 females constituted the pre-NBS group. The median age (range) of the NBS and pre-NBS groups was 28.5 years (20.5 to 38.2) and 43.9 years (37.7 to 50.8), respectively. Regarding the clinical form of PKU, almost all patients had classic PKU, but six patients with mild HPA and two patients with BH_4_-responsive PKU were included in the NBS group. In this study, almost all participants (66 of the 68 patients in the NBS group and 16 of the 17 patients in the pre-NBS group) were managed by pediatricians even after reaching adulthood.

Approximately 30% of patients (21 of the 68 patients in the NBS group and 5 of the 17 patients in the pre-NBS group) had ever discontinued treatment. Although the description of treatment discontinuation was unequal due to the open-ended questionnaire, concrete information was obtained for only eight patients in the NBS group. Among them, the median age at treatment discontinuation was 18 years (range, 11 to 20 years), and the median duration was 11 years (range, a few years to 23 years). In the pre-NBS group, the details of treatment discontinuation were provided by one patient, and she was a 42-year-old female who had several intermittent histories of treatment discontinuation for a total of a few decades beginning at 3 years of age. As shown in Table 2, the most common reasons for discontinuation before and after NBS were financial problems and self-judgment, including a decrease in motivation and busy school or work schedules, followed by the unpleasant taste of the special formula and erroneous recommendations from the attending physicians. On the other hand, the most common reason for restarting dietary therapy is the prevention of maternal PKU, followed by improvement of the medical subsidy system and the appearance of psychiatric or behavioral abnormalities.

Figure 1 shows the comparison of intellectual outcomes before and after NBS in different age groups. Normal intelligence was observed in 60 of the 68 patients (88%) in the NBS group; specifically, all but two patients less than 35 years old had normal intelligence. Meanwhile, even in the NBS group, 6 of 13 patients over 35 years of age exhibited a certain degree of intellectual disability, including 5 patients with borderline intellectual disability. In contrast to the results of the NBS group, only 6 of the 17 patients (35%) in the pre-NBS group showed normal intelligence. Additionally, the degree of intellectual disability in the pre-NBS group was more severe than that in the NBS group.

Table 3 shows a comparison of other clinical symptoms and social outcomes. Regarding the psychiatric status, transient psychiatric disabilities were observed in three patients in the NBS group, all of whom had histories of treatment interruption, during their intermittent treatment. Another patient presented with a psychiatric disability. Four patients presented with an abnormal psychiatric status in the NBS group (6%). On the other hand, 6 of 17 patients (35%) in the pre-NBS group had psychiatric disabilities. When the number of patients with intellectual and/or psychiatric disabilities was counted collectively as neuropsychiatric diseases, the total was 10 patients in the NBS group. More specifically, six patients were diagnosed with only an intellectual disability, two patients were diagnosed with intellectual and psychiatric disabilities, and two patients were diagnosed with only a transient psychiatric impairment during treatment interruption. Meanwhile, 12 patients in the pre-NBS group were diagnosed with neuropsychiatric diseases (six patients with only intellectual disability, five with intellectual and psychiatric disabilities, and one with only psychiatric disability).

The relationship between neuropsychiatric symptoms and treatment interruption is described below. In the NBS group, 5 of 21 patients with treatment interruption had a neuropsychiatric disease, while 5 of 47 patients who were continuing treatment had a neuropsychiatric disease. A 35-year-old female patient who had the earliest and longest treatment discontinuation period (for approximately 23 years beginning at an age of 11 years) in the NBS group presented with a psychiatric disability and “moderate-severe intellectual disability” that was the worst intellectual level in the NBS group. Although 5 of 10 patients in the NBS group with a neuropsychiatric disability had a history of treatment discontinuation, the details, such as how long and when treatment discontinued, and Phe levels, were not accurately obtained. On the other hand, in the pre-NBS group, four of five patients with treatment interruption had a neuropsychiatric disease, while 8 of 12 patients who were continuing treatment had a neuropsychiatric disease. From another perspective, 4 of 12 patients with a neuropsychiatric disability had a history of treatment discontinuation. A 42-year-old female in the pre-NBS group who discontinued treatment at 3 years old based on the recommendation of an attending physician, as mentioned above, was unable to be employed, despite her normal intelligence, due to a psychiatric disability induced by intermittent treatment discontinuation for a few decades.

The rate of normal physical size in the NBS group was higher than that in the pre-NBS group, but no obvious differences in physical measurements, including short stature, obesity, or leanness, were observed between the two groups.

Information on education status was obtained for 57 of the 68 patients in the NBS group and 12 of the 17 patients in the pre-NBS group. Fifty-five of 57 (96%) patients in the NBS group had an education level of high school or higher, while 7/12 (58%) patients in the pre-NBS group graduated from high school or higher educational institutions.

Regarding employment status, all but two patients (97%) in the NBS group were employed or attended universities. One of the two unemployed patients was a married female with normal intelligence and was managed to prevent maternal PKU. She was likely a full-time housewife, although this information was not provided. Meanwhile, seven patients (41%) in the pre-NBS group were employed. Five patients each in the pre-NBS group were unemployed or were living in a facility for individuals with a disability. Four of five unemployed patients had intellectual and/or psychiatric disabilities. The other unemployed patient had normal intelligence but also had a tracheostomy due to other respiratory diseases. In summary, all but one of the patients in the NBS group was able to live an independent life, while over half of the patients in the pre-NBS group were not employed or lived in facilities for individuals with disabilities.

Regarding the marital status, 19 of 50 patients in the NBS group were married, while 2 of 15 patients in the pre-NBS group were married.

When given the opportunity to freely describe the patient, many attending physicians stated the difficulty in continuing a Phe-restricted diet, the necessity for lifelong treatment, the need to support medical expenses, and/or the necessity for consultation with internal medicine and psychiatric specialists.

## 4. Discussion

Our study examined the physical, neurological and social status of adult patients who were diagnosed with PKU before and after NBS in Japan and revealed that NBS obviously contributed to the improvement of long-term outcomes; our results were similar to those of previous reports [1,15,16,17]. While almost all patients in the NBS group exhibited normal mental development and were able to live an independent life, over half of the patients in the pre-NBS group had neurological and psychiatric problems and were more severely disabled. However, even in the NBS group, 10 patients had intellectual and/or psychiatric disabilities. Because five of them had histories of treatment interruption, continuous and lifelong treatment is essential for good neurological outcomes.

Meanwhile, five patients in the NBS group whose disease remained well-controlled from birth presented with intellectual and/or psychiatric disabilities despite having no history of treatment interruption. In particular, the neurological outcomes of patients in their late 30s (born from 1977 to 1981) were not as good as those in the younger age group. Four of five patients with neuropsychiatric diseases despite continuing the treatment were in their late 30s. This difference is due to less strict target levels of Phe at the beginning stage of NBS [3,18]. In fact, the target levels of plasma Phe concentration were 4–8 mg/dL in babyhood and 4–12 mg/dL in childhood in Japanese guidelines at that time. In addition, because the recommended Phe levels after the age of 6 years were not indicated until the early 1990s, some Japanese physicians considered that dietary therapy could be relaxed and the dietary restriction could be discontinued after patients reached school age. Because the restriction of Phe was not sufficient even before 20 years, a patient aged in the early 20s had a borderline intellectual disability, despite having no history of poor Phe management. As well as this patient, patients who were detected by NBS, even those with proper management, showed poor neurological outcomes in previous studies [16,19]. Hence, in the present guidelines, the restriction of Phe has become stricter. Namely, not only continuous but also more restricted diet therapy is necessary for good neurological outcomes.

Normal intelligence was observed in some patients in the pre-NBS group despite treatment after onset for unexplained reasons. Additionally, some untreated individuals with classic PKU have normal intelligence despite the elevated plasma Phe concentration [1]. However, the reasons are still unknown. In our study, the symptoms at the onset, what triggered their diagnosis, and when and what type of treatment they had received were not recorded, while our results revealed that their clinical forms were all classic, their age ranged from 39 to 46 years, and one patient had a treatment interruption for 7 years, indicating that the factors constituting a good neurological prognosis are unknown. 

Our study also revealed the reasons why patients with PKU discontinued and restarted treatment. Although the unpleasant taste of the special formula is well known [20], treatment interruption was more frequently caused by economic problems in our study. In fact, because the medical public support system for adult patients with PKU has improved since 2015 in Japan, some patients were able to restart treatment. Therefore, lifelong administrative-economic support is necessary to continue treatment. Furthermore, treatment was often neglected due to the self-judgment of being free of symptoms and personal circumstances, such as busy school life and working during adulthood or after childhood. Although many patients detected by Japanese NBS are strictly controlled in childhood by physicians, nutritionists, medical staff, and family, they are released from strict management after adulthood due to independence from those supporters. Therefore, adult patients tend to neglect regular visits and strict dietary therapy due to poor subjective symptoms, and neurological symptoms progress over a chronic course. On the other hand, the primary reason for restarting treatment was pregnancy and the prevention of maternal PKU. Therefore, we strongly suggest that appropriate and continuous patient education is the most effective method to prevent loss to follow-up.

In our study, psychiatric disabilities were likely to progress in the later period of life. One of the reasons may be a difference in the quality of disease management between childhood and adulthood, as mentioned above. In fact, some patients present with psychiatric disabilities after adulthood due to inadequate self-management or treatment intermittence despite having normal intelligence and receiving higher education in childhood. Additionally, it was previously reported that neuropsychiatric symptoms are more prevalent in older adults with PKU [21]. Namely, PKU is considered a progressive disease even after adulthood. Higher blood levels of Phe inhibit myelination in early childhood and functionally impair myelin in late childhood or adulthood, even after normal myelination [22]. Thus, continuous treatment is important for favorable neuropsychiatric outcomes.

Regarding physical development, no obvious differences were detected between the NBS and pre-NBS groups. Nevertheless, optimal growth outcomes were not attained in a previous study, even with advances in dietary treatments [23]. Although our study did not indicate major issues related to growth, physical development should be continuously evaluated for each individual with PKU.

Our results also indicated that the transition to an adult internal medicine department is not easy at this time in Japan. In fact, this issue has been observed in not only Japan but also other countries [24] and for not only PKU patients but also patients with other rare diseases [25]. The explanation for this finding is that internal medicine physicians for adults have little knowledge of PKU or are unable to appropriately treat this disease. In addition and very importantly, patients with PKU realize this limitation and generally prefer to be treated by pediatricians who are experts in PKU care and treatment. Furthermore, patients feel comfortable visiting a place where they know the care providers rather than a new and strange facility. Therefore, the transition to adult department is a major issue for all adult patients with PKU. Furthermore, the issue of the transition and when to make transition have not been described yet in Japanese guidelines, although preparations for the transition from pediatrics to adult internal medicine should begin at approximately 12 years of age based on European guidelines [18,26]. Namely, transitions are likely to be delayed in Japan compared with Europe. 

Finally, our study may have some limitations. Only patients who were treated with Phe-free formula and could be traced after adulthood were recruited for our study. According to the annual report of Japanese NBS, approximately 400 adult patients were estimated to be diagnosed with PKU during the period from 1977 to 1996. However, in the present study, only 68 patients were enrolled in the NBS group. Our study did not include a considerable number of adult patients with PKU who no longer received treatment at the hospital, discontinued treatment with Phe-free formula, were treated with only BH_4_, and died before adulthood. Furthermore, because the participants in this study had continuously or intermittently ordered the special formula, they were willing to be treated even after reaching adulthood. Therefore, our results might be shifted to better outcomes of PKU than the actual situation. Nevertheless, our results are useful for many Asian physicians because the long-term outcomes of East Asian patients with PKU have not been well understood to date. Improvements in the patient registration, follow-up and medical support systems for adult and pediatric patients with PKU will be essential for achieving lifelong favorable outcomes.

## 5. Conclusions

The long-term outcomes of adult patients with PKU detected by NBS were much more favorable than those of patients in the pre-NBS group in Japan, as previously reported in other countries. However, some patients, even those who underwent early detection using NBS, with histories of treatment intermittence suffered from neuropsychiatric symptoms. Lifelong and strict management is essential to maintain a good prognosis for patients with PKU.

## Figures and Tables

**Figure 1 IJNS-07-00021-f001:**
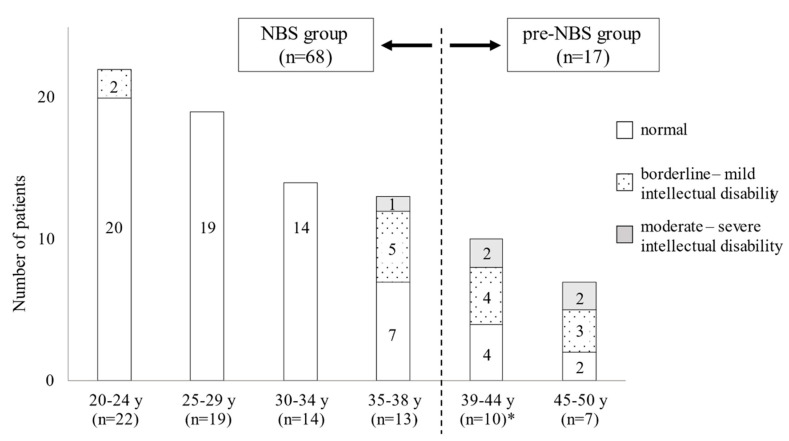
Degree of intellectual disability in patients stratified by age before and after newborn screening.*, A 37-year-old female was included in the “39–44-year-old subgroup” of the pre-newborn screening (NBS) group because she did not undergo NBS.

**Table 1 IJNS-07-00021-t001:** Overview of the participating patients with phenylketonuria (PKU).

		NBS Group (*n* = 68)	Pre-NBS Group (*n* = 17)
Sex			
	male	34	7
	female	33	10
	unknown	1	0
Median age (range) [years]		28.5 (20.5–38.2)	43.9 (37.7–50.8)
	20–24	22	0
	25–29	19	0
	30–34	14	0
	35–39	13	3
	40–44	0	7
	45–49	0	6
	50-	0	1
Clinical form			
	classic	54	16
	mild HPA	6	0
	BH_4_-responsive	2	0
	BH_4_ defect	0	0
	unknown	6	1
Follow-up department			
	pediatrics	66	16
	internal medicine	1	1
	gynecology	1	0
treatment interruption		21	5

HPA, hyperphenylalaninemia; BH_4_, tetrahydrobiopterin.

**Table 2 IJNS-07-00021-t002:** Reasons for discontinuing or restarting dietary treatment.

Reasons for Discontinuation (Number of Patients with the Same Comment)	Reasons for Restarting (Number of Patients with the Same Comment)
Economic problems (7) Self-judgment/personal circumstances (7) Unpleasant taste of the special formula (6) Recommendation of the attending physicians (5) Changes in one’s environment (3)	Pregnancy (8) Improvement of the medical subsidy system (4) Appearance of psychiatric abnormalities (3) Spontaneously restarted (2)

**Table 3 IJNS-07-00021-t003:** Comparison of clinical and social parameters.

	NBS Group (*n* = 68)					Pre-NBS Group (*n* = 17)		
	20–24 y (*n* = 22)	25–29 y (*n* = 19)	30–34 y (*n* = 14)	35–38 y (*n* = 13)	Total (%)	39–44 (*n* = 10) *	45–50 y (*n* = 7)	Total (%)
Psychiatric status								
normal	20	19	13	12	64 (94%)	7	4	11 (65%)
transient impairment during treatment interruption	2	0	1	0	3 (4%)	0	0	0 (0%)
psychiatric disability	0	0	0	1	1 (1%)	3	3	6 (35%)
Physical characteristics								
normal	19	13	14	11	57 (84%)	5	7	12 (71%)
short stature	1	0	0	0	1 (1%)	1	0	1 (6%)
obesity	2	3	0	1	6 (9%)	2	0	2 (12%)
obesity and short stature	0	2	0	0	2 (3%)	0	0	0 (0%)
leanness	0	1	0	1	2 (3%)	1	0	1 (6%)
unknown	0	0	0	0	0 (0%)	1	0	1 (6%)
Education status								
university	9	10	3	4	26 (38%)	1	1	2 (12%)
technical school	3	5	3	0	11 (16%)	1	0	1 (6%)
high school ^#^	10	2	4	2	18 (26%)	2	2	4 (24%)
junior high school	0	0	0	1	1 (1%)	0	0	0 (0%)
school for individuals with a disability	0	0	0	1	1 (1%)	3	2	5 (30%)
unknown	0	2	4	5	11 (16%)	3	2	5 (30%)
Employment status								
attending school	4	1	0	0	5 (7%)	0	0	0 (0%)
employed	18	17	14	12	61 (90%)	3	4	7 (41%)
unemployed	0	1	0	1	2 (3%)	4	1	5 (30%)
living in a house for individuals with a disability	0	0	0	0	0 (0%)	3	2	5 (30%)
unknown	0	0	0	0	0 (0%)	0	0	0 (0%)
Marital status								
married	0	4	9	6	19 (28%)	2	0	2 (12%)
unmarried or divorce	14	8	4	5	31 (46%)	8	5	13 (76%)
unknown	8	7	1	2	18 (26%)	0	2	2 (12%)

Short stature was defined as a height <160 cm for males and <148 cm for females. Obesity and leanness were defined as BMI >25 and <18.5, respectively. *, A 37-year-old female is included in the group of “39–44 years old” because she did not undergo NBS. ^#^, Graduation from “high school” includes five patients who are still in college.

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
