# Peer review of "Long-Term Neurological Outcomes of Adult Patients with Phenylketonuria before and after Newborn Screening in Japan"

_2409-515X, 2021, doi:10.3390/ijns7020021_

Round 1

Reviewer 1 Report

Introduction:

Line 32: what is a representative disease? I do not think that is a correct use of the word representative.

Please provide MIM and gene number for the disease, enzyme and gene.

Treatment of PKU is stated to be dietary with Phe-free formula. This should be nuanced: a complete Phe free diet is not advised as this can lead to other problems. It should be stated that it is a diet with protein restriction and Phe-free formula (some Phe is necessary, also in PKU!).

Line 48 and 59: remove - in the words prognoses and questionnaire. Same for the word 'included' line 68. Please check manuscript for other spelling mistakes.

METHODS

Line 70, please add for the lengt what standard deviation scores these are.

RESULTS

The results are to forward for a descriptive study.

For example, reasons for discontinuation of treatment are given. This is relevant, but more interesting would be to know how long discontinuation was in these patients, what there intelligence is, what period and at what age the discontinuation occured etc. This would be a very interesting outcome of this study.

I think more details can be given since it is a small group of patients. 

CONCLUSION

Very interesting in depth discussion.

One advice: perhaps it would be good to take a glance at the part where you first state that some post NBS patients with an intellectual deficit did not have treatment discontinuation. However you go on to hypothesizing that it might be because of faulty doctors advice or because of not going to check-ups etc. To me this does constitute discontinuation of treatment. Please rewrite this section appropriatly.

Author Response

Thank you for your review and constructive comments. We are honored that you appreciate our manuscript. Our manuscript has been proofread again by native English-speakers, and all changes are shown in red.

Introduction:

Line 32: what is a representative disease? I do not think that is a correct use of the word representative.

According to your suggestion, we changed “representative” to “major”.

Please provide MIM and gene number for the disease, enzyme and gene.

We added the OMIM numbers of PKU and PAH.

Treatment of PKU is stated to be dietary with Phe-free formula. This should be nuanced: a complete Phe free diet is not advised as this can lead to other problems. It should be stated that it is a diet with protein restriction and Phe-free formula (some Phe is necessary, also in PKU!).

We agree with your comments and changed this point according to your suggestion.

Line 48 and 59: remove - in the words prognoses and questionnaire. Same for the word 'included' line 68. Please check manuscript for other spelling mistakes.

We apologize but we cannot remove the “hyphens”. When our document file (made by Microsoft “Word”) is converted to a PDF file, hyphens are automatically and randomly inserted. Moreover, we cannot revise this PDF file because we cannot check this file before our submission.

METHODS

Line 70, please add for the lengt what standard deviation scores these are.

We apologize but we do not understand your comment. What does “length what standard deviation scores” mean? We did not calculate any SD in our study because our data are mainly descriptive and not numerical values. Therefore, we did not address this comment.

RESULTS

The results are to forward for a descriptive study.

For example, reasons for discontinuation of treatment are given. This is relevant, but more interesting would be to know how long discontinuation was in these patients, what there intelligence is, what period and at what age the discontinuation occured etc. This would be a very interesting outcome of this study.

I think more details can be given since it is a small group of patients. 

Thank you for your constructive comments. However, the detailed characteristics of some patients were not provided and were unequal because of the open-ended question. We added the details of the duration and age at treatment interruption as much as possible.

CONCLUSION

Very interesting in depth discussion.

One advice: perhaps it would be good to take a glance at the part where you first state that some post NBS patients with an intellectual deficit did not have treatment discontinuation. However you go on to hypothesizing that it might be because of faulty doctors advice or because of not going to check-ups etc. To me this does constitute discontinuation of treatment. Please rewrite this section appropriatly.

According to your suggestions, we changed the 2nd paragraph of the Discussion. 

Reviewer 2 Report

This paper provides useful outcome information on patients with PKU identified in Japan before and after NBS.

Lines 105-107 and 233-241   It is not surprising that almost all adult patients continue to be managed by pediatric facilities rather than in adult settings. Physicians who care for adults rarely know about PKU or can appropriately treat them. This is true throughout the world. In addition and very importantly, patients with PKU realize this and much prefer to continue going to pediatric centers that are expert in PKU care and treatment. Furthermore, they feel comfortable going to a place where they know the care providers rather than to a new and strange facility. This should be elaborated upon in Discussion.

Lines 164-165    One patient in the pre-NBS group has normal intelligence. We need more information about this patient.  What symptoms led to her diagnosis of PKU? At what age was she diagnosed? When did PKU treatment begin? Has PKU treatment been continuous since it began?

Rare patients with PKU who have never been treated or who were late treated yet have normal intelligence have been reported . I suggest that a paragraph on this be added to Discussion.

Author Response

This paper provides useful outcome information on patients with PKU identified in Japan before and after NBS.

Thank you for your review and constructive comments. We are honored that you appreciate our manuscript. Our manuscript has been proofread again by native English-speakers, and all changes are shown in red.

Lines 105-107 and 233-241   It is not surprising that almost all adult patients continue to be managed by pediatric facilities rather than in adult settings. Physicians who care for adults rarely know about PKU or can appropriately treat them. This is true throughout the world. In addition and very importantly, patients with PKU realize this and much prefer to continue going to pediatric centers that are expert in PKU care and treatment. Furthermore, they feel comfortable going to a place where they know the care providers rather than to a new and strange facility. This should be elaborated upon in Discussion.

We agree with your comments and incorporated your suggestions in the Discussion.

Lines 164-165    One patient in the pre-NBS group has normal intelligence. We need more information about this patient.  What symptoms led to her diagnosis of PKU? At what age was she diagnosed? When did PKU treatment begin? Has PKU treatment been continuous since it began?

We did not collect this information. Although we were unable to add this information, we expanded the discussion of patients with PKU who had normal intelligence despite starting treatment after onset in the 3rd paragraph of the Discussion.

Rare patients with PKU who have never been treated or who were late treated yet have normal intelligence have been reported . I suggest that a paragraph on this be added to Discussion.

According to your suggestions, we added a paragraph describing patients with PKU who have normal intelligence despite late treatment as mentioned above. However, we minimized the length of the description because our data do not include this information.

Round 2

Reviewer 1 Report

Much better.